# GENERATIVE REWARD MODELS

## ABSTRACT

Reinforcement Learning from Human Feedback (RLHF) has greatly improved the performance of modern Large Language Models (LLMs). The RLHF process is resource-intensive and technically challenging, generally requiring a large collection of human preference labels over model-generated outputs. Reinforcement Learning from AI Feedback (RLAIF) addresses this data collection challenge by leveraging synthetic preferences generated by an LLM. However, recent work has shown that synthetic preferences labels may not align well with human preference judgments (Zeng et al., 2023). To address this, we propose a hybrid approach that unifies RLHF and RLAIF methodologies. We introduce **GenRM**, an iterative algorithm that trains an LLM on self-generated reasoning traces, leading to synthetic preference labels matching human preference judgments. Empirically, we show that zero-shot LLM-based judgments under-perform compared to Bradley-Terry reward models on in-distribution tasks (between 9-36%). In contrast, GenRM achieves in-distribution accuracy comparable to Bradley-Terry models, while significantly outperforming them on out-of-distribution tasks (between 10-45%). Moreover, GenRM surpasses the performance of using LLMs as judges on both in-distribution (by 9-31%) and out-of-distribution tasks (by 2-6%). Our results show that combining the strengths of RLHF and RLAIF offers a promising approach for improving the quality of synthetic preference labels.

## 1 INTRODUCTION

Reinforcement Learning from Human Feedback (RLHF) has significantly improved the performance of modern Large Language Models (LLMs) (see e.g., Reid et al., 2024; OpenAI, 2023). Despite its effectiveness, the RLHF process presents several challenges. First, it requires a large amount of human preference data to train reward models that reflect human preferences (Stiennon et al., 2022; Bai et al., 2022a). Second, it necessitates additional architecture and infrastructure to handle reward model training (Wang et al., 2024a; von Werra et al., 2020; Havrilla et al., 2023). Third, it requires a sophisticated online optimization loop using algorithms, such as Proximal Policy Optimization [PPO; Schulman et al. (2017)], to fine-tune an LLM-based policy to align with the reward model (Zheng et al., 2023c).

To address the challenge of collecting large-scale human preference data, synthetic preference data has emerged as a promising alternative. For example, Bai et al. (2022b) introduced Reinforcement Learning from AI Feedback (RLAIF). Instead of relying on human users for feedback, their method utilizes an LLM guided by a predefined set of principles—referred to as a "constitution"—to generate and select model outputs that are helpful and harmless (Askell et al., 2021). Employing AI-generated preference labels has demonstrated meaningful Pareto improvements in balancing helpfulness and harmlessness in assistant responses (Bai et al., 2022b; Kundu et al., 2023).

Direct alignment algorithms, such as Direct Preference Optimization (DPO) (Rafailov et al., 2023) and Implicit Preference Optimization (IPO) (Azar et al., 2023), were developed to address the challenges of reward model training and online optimization. These works demonstrated that the reward model and the optimal policy can be mathematically interchanged, allowing the policy to be trained directly from preference data in an entirely offline manner, significantly simplifying the RLHF pipeline. Benchmark evaluations (Lambert et al., 2024) have shown that DPO-based approaches are competitive with traditional reward models based on the Bradley-Terry algorithm. However, recent empirical evidence suggests that purely offline methods may underperform compared to online approaches in both reward model-based reinforcement learning (Xu et al., 2024b;a) and in the

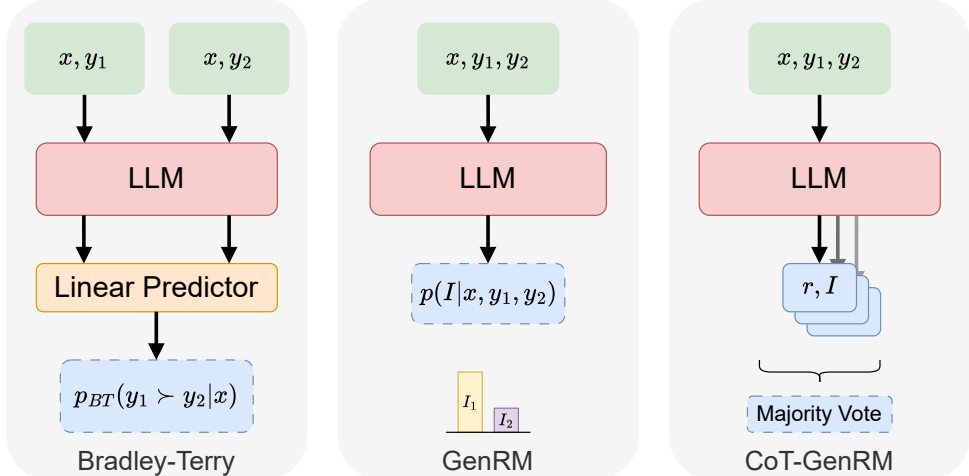

Figure 1: **Methods overview.** *Bradley-Terry* methods directly output the probability of $y_1$ being preferred over $y_2$, while *GenRM* compares the LLMs next-token probabilities of answer indicator tokens ($I_1, I_2$). *CoT-GenRM* samples reasoning traces ($r$) followed by the answer indicator token.

RLAIF setting (Guo et al., 2024). As a result, state-of-the-art models such as the LLaMA-3 family (Dubey et al., 2024) have adopted hybrid strategies that combine online DPO optimization with separate reward models.

In this work, we identify two key **limitations** in current alignment approaches: (1) Explicitly parameterized reward models, while effective and accurate for in-distribution tasks, struggle with robustness and generalization to out-of-distribution (OOD) data. (2) RLAIF approaches, such as utilizing an LLM-as-a-judge, offer a more robust alternative but may not always align well with actual user preferences when acting as the sole evaluator. To **address these limitations**, we propose a unified framework for RLHF and RLAIF. Our approach begins with a strong pre-trained LLM, which we employ as an evaluator. Using a dataset of user preferences, we adopt a STaR-like methodology (Zelikman et al., 2022) to align the LLM with user choices, effectively training it to function as a reward model. We **demonstrate empirically** that this fine-tuned judge model matches Bradley-Terry reward models for in-distribution prompts while significantly improving generalization on OOD prompts. Additionally, it outperforms the base LLM on both in-distribution and OOD scenarios.

## 2 PRELIMINARIES

In this section, we first outline the core components of the standard RLHF post-training pipeline (Ziegler et al., 2020; Stiennon et al., 2022; Bai et al., 2022a; Ouyang et al., 2022), then summarize the Self-Taught Reasoner (STaR) approach (Zelikman et al., 2022), and finally review LLM-as-a-judge (Zheng et al., 2023a).

### 2.1 REINFORCEMENT LEARNING FROM HUMAN FEEDBACK

The RLHF pipeline consists of three stages designed to align an LLM with human preferences: (1) Supervised finetuning (SFT); (2) Reward Modeling; and (3) Reinforcement Learning (RL).

#### 2.1.1 SUPERVISED FINE-TUNING (SFT)

In the first stage, an LLM is trained to follow instructions using a dataset of prompts $x$ and responses $y$ using maximum likelihood estimation (MLE) over the next-token predictions. The resulting model is referred to as $\pi_{\text{SFT}}(y \mid x)$, where both the prompt and response strings are treated as single variables. This model is used as a base for the next stages.

### 2.1.2 Bradley-Terry Reward Modeling

Next, the SFT model $\pi_{\text{SFT}}(y \mid x)$ is leveraged to construct a reward model that captures human preferences. Specifically, the SFT model is sampled, generating pairs of responses $(y_1, y_2) \sim \pi_{\text{SFT}}(y \mid x)$ for each prompt $x$ in the dataset. Human annotators then rank the responses, producing pairs of preferences $y_w \succ y_l \mid x$, where $y_w$ and $y_l$ represent the preferred and non-preferred responses, respectively. This ranking process is typically modeled using the Bradley-Terry (BT) preference model (Bradley & Terry, 1952), which assumes the preference distribution:

$$p_{\text{BT}}(y_1 \succ y_2 \mid x) = \frac{\exp\left(r(x, y_1)\right)}{\exp\left(r(x, y_1)\right) + \exp\left(r(x, y_2)\right)} = \sigma\left(r(x, y_1) - r(x, y_2)\right), \tag{1}$$

where the preference distribution $p$ is driven by a latent reward function $r(x, y)$, and $\sigma$ is the logistic function (although other objectives can be used). Using this framework and a dataset of rankings $\mathcal{D} = \left\{x^{(i)}, y_w^{(i)}, y_l^{(i)}\right\}_{i=1}^{N}$, a parameterized reward model $r_\phi(x, y)$ is trained via maximum likelihood estimation to predict the unobserved reward:

$$\mathcal{L}_{\text{rew}}(r_\phi) = -\mathbb{E}_{(x, y_w, y_l) \sim \mathcal{D}} \left[\log \sigma\left(r_\phi(x, y_w) - r_\phi(x, y_l)\right)\right]. \tag{2}$$

This reward model is based on $\pi_{\text{SFT}}(y \mid x)$ with an additional linear predictor on top of the final embedding layer of the model which produces the scalar reward estimate.

### 2.1.3 Reinforcement Learning (RL)

In the final stage, the learned reward model $r_\phi(x, y)$ is used to further optimize the LLM $\pi_\phi$ via an on-policy RL algorithm, such as Proximal Policy Optimization (PPO; Schulman et al., 2017). The goal is to refine the LLM's behavior so that it produces responses preferred by human evaluators. The common optimization objective is:

$$\max_{\pi_\theta} \mathbb{E}_{x \sim \mathcal{D}, y \sim \pi_\phi(.|x)} \left[r_\phi(x, y) - \beta \mathbb{D}_{\text{KL}}\left[\pi_\theta(y \mid x) \parallel \pi_{\text{ref}}(y \mid x)\right]\right], \tag{3}$$

where $\mathbb{D}_{\text{KL}}$ represents the Kullback-Leibler (KL) divergence, and $\pi_{\text{ref}}(y \mid x)$ is typically the supervised fine-tuned model $\pi_{\text{SFT}}(y \mid x)$. The KL divergence penalty prevents the LLM $\pi_\phi$ from deviating too far from its initial behavior, with the hyperparameter $\beta$ controlling the trade-off between exploiting the reward model and maintaining consistency with the reference model.

## 2.2 Self-Taught Reasoner

The Self-Taught Reasoner (STaR) method introduces an iterative bootstrapping approach designed to improve the reasoning capabilities of LLMs (Zelikman et al., 2022). STaR focuses on training models to generate and refine rationales, particularly for tasks requiring complex reasoning in a reinforcement learning-based manner. We outline the main points of the approach below.

### 2.2.1 Rationale Generation Bootstrapping

In our formulation we assume we have access to a dataset $\mathcal{D} = \{x^{(i)}, y^{(i)}\}_{i=1}^{N}$ of questions $x$ that require strong reasoning and the corresponding answers $y$. Notice that we do not require access to strong or ground-truth rationales for these problems. We begin by prompting a model $\hat{y}^{(i)}, \hat{r}^{(i)} \sim \pi(y, r|x^{(i)})$ to provide CoT rationale $\hat{r}^{(i)}$ and final answer $\hat{y}^{(i)}$. We then filter the generated data, keeping only rationales leading to a correct final answer (i.e. $\hat{y}^{(i)} = y^{(i)}$) to generate a dataset of questions, (bootstrapped) rationales and answers $\mathcal{D}_{\text{STaR}} = \{x^{(i)}, \hat{r}^{(i)}, y^{(i)}\}_{i=1}^{N}$. $\mathcal{D}_{\text{STaR}}$ is then used to train a model with the standard supervised fine-tuning objective:

$$\mathcal{L}_{\text{STaR}}(\pi_\phi) = -\mathbb{E}_{(x, \hat{r}, y) \sim \mathcal{D}_{\text{STaR}}} \left[-\log \pi_\phi(y, \hat{r}|x)\right]. \tag{4}$$

The above procedure is repeated over several iterations and has since been adopted in various related works (e.g., Hosseini et al., 2024; Andukuri et al., 2024; Fränken et al., 2024; Zelikman et al., 2024).

### 2.2.2 POST-RATIONALIZATION

One limitation of bootstrapping rationale generation is that the model cannot improve on examples it initially fails to solve. To address this issue, STaR introduces *rationalization*, a backward reasoning process. For prompts where the model generates an incorrect rationale and answer, the correct answer is provided to the model as a hint. The model then generates a new rationale based on the correct answer, reasoning backward to generate a "post-rationale". In technical terms there is a rationalization model $q$ which generates rationales $\hat{r}^{(i)} \sim q(r|x^{(i)}, y^{(i)})$ to justify the final answer. This synthetic data is in turn used in the STaR objective in Eq. 4. We will also use this approach to evaluate the effect of the quality of the bootstrapped reasoning chains, using samples from rationalization models $q$ with different capabilities.

### 2.3 RLAIF AND LLM-AS-A-JUDGE

Reinforcement Learning from AI Feedback (RLAIF) presents an alternative approach to the standard RLHF pipeline. Bai et al. (2022b) demonstrate the efficacy of RLAIF in training helpful and harmless models without relying on human feedback labels for harmlessness assessment. Their work shows that as language model capabilities improve, AI identification of harms increases significantly, particularly when leveraging chain-of-thought reasoning. Notably, they demonstrate that utilizing self-supervised preference labels for reinforcement learning can yield improvements in model behavior that are competitive with or surpass those achieved using human feedback for harmlessness evaluation. Zheng et al. (2023a) introduce the LLM-as-a-Judge method, further extending the RLAIF paradigm. They demonstrate that strong language models, even without explicit training for evaluation tasks, can provide judgments that exhibit agreement with human preferences. Their study finds that LLMs can achieve over 80% agreement with human preferences, a level comparable to inter-expert agreement. This finding establishes a foundation for developing LLM-based evaluation frameworks.

## 3 CONNECTIONS BETWEEN RLHF AND PREFERENCE MODELLING

We begin by pointing out a theoretical connection between the RLHF post-training approach outlined in the previous section and general preference modeling. One of the key observations of the DPO approach (Rafailov et al., 2023) is that the reward modeling objective in Eq. 1 is underconstrained, which can create significant optimization challenges for the RL problem in Eq. 3 (Wu et al., 2023; Ahmadian et al., 2024) (in fact, theoretically, it can have arbitrarily low signal-to-noise ratio). To alleviate this issue, prior works (Stiennon et al., 2022; Ouyang et al., 2022) use a baseline reward from on a fixed reference distribution:

$$\max_{\pi_\theta} \mathbb{E}_{x \sim \mathcal{D}, y \sim \pi_\phi(.|x)} \Big[ [r_\phi(x, y) - r_\phi(x, y_{\text{ref}})] - \beta \mathbb{D}_{\text{KL}} \left[ \pi_\theta(y \mid x) \parallel \pi_{\text{ref}}(y \mid x) \right] \Big]. \tag{5}$$

Here the reference $y_{\text{ref}}$ is a human completion from an SFT dataset or a sample from the base SFT model. However, notice that by inverting Eq. 1 we have:

$$r_\phi(x, y) - r_\phi(x, y_{\text{ref}}) = \log \left( \frac{p_{\text{BT}}(y \succ y_{\text{ref}}|x)}{1 - p_{\text{BT}}(y \succ y_{\text{ref}}|x)} \right). \tag{6}$$

That is, under the Bradley-Terry formulation the standard RLHF optimization procedure is actually optimizing a preference likelihood objective.

The **core contribution** of our work is that we replace the Bradley-Terry reward modelling approach with a strictly more general preference modelling objective $p(y_w \succ y_l|x)$ that does not assume a point-wise reward estimate, a special model architecture, or a specific preference distribution parameterization as in Eq. 1. That is, we assume a standard preference dataset and model the preference distribution $p_\phi(y_w \succ y_l|x)$ using an LLM, without any additional assumptions. Notice that this formulation is fully general as we can extract preference probabilities either from the likelihoods of the LLM output or from majority voting counts. This can be used in the standard pipeline with PPO (Stiennon et al., 2022; Ouyang et al., 2022) in Eq. 5 using the reward formulation in Eq. 6. Alternatively, if we sample preferences from the above model, we can also utilize an iterative or online *PO optimization manner following Munos et al. (2024) or Calandriello et al. (2024).

# 4 GENERATIVE REWARD MODELS: A UNIFIED RLHF-RLAIF APPROACH

In our proposed framework we begin with an LLM $p_\phi$ (we will use the notation $p_\theta$ instead of $\pi_\theta$ to highlight that we are referring to an evaluation model rather than the policy itself) acting as a zero-shot judge in an RLAIF setting, as outlined in Section 2.3. That is, given a task $x$ and two responses $y_1$ and $y_2$, we directly prompt the model $p_\phi$ to provide an answer indicator token $I$ indicating a preference over the answers. We consider two variants of our approach:

1. The Generative Reward Model (GenRM) approach prompts the model to act as a classifier directly providing the answer token probabilities for each response $\hat{I} \sim p_\phi(I|x, y_1, y_2)$.

2. The CoT-GenRM approach additionally prompts the model to provide intermediate Chain-of-Thought reasoning $\hat{I}, \hat{r} \sim p_\phi(I, r|x, y_1, y_2)$ before providing the final answer indicator token.

Our prompts are based on the standard MT-Bench prompt (Zheng et al., 2023b), and can be found in Appendix B. We use the LLM judge as a prior and further train it to align with the ground-truth dataset judgements. We begin with the preference dataset $\mathcal{D} = \left\{ x^{(i)}, y_1^{(i)}, y_2^{(i)}, I^{(i)} \right\}_{i=1}^N$ as outlined in Section 2.1.2, however we consider unranked answers $y_1^{(i)}, y_2^{(i)}$ and the corresponding winning choice $I^{(i)}$. We design several training techniques for the generative reward model $p_\phi$.

**GenRM (no CoT):** To train the GenRM model, we use the standard supervised fine tuning objective

$$\mathcal{L}_{\text{GenRM}}(\pi_\phi) = \mathbb{E}_{(x, y_1, y_2, I) \sim \mathcal{D}}[-\log p_\phi(I|x, y_1, y_2)] \tag{7}$$

essentially using the LLM as a classifier trained with next-token prediction.

**CoT-GenRM with Rationalization:** To train the CoT-GenRM we will also consider two settings—bootstrapping intermediate reasoning from ground-truth or, potentially, a stronger rationalization model $r \sim q_\phi(r|x^{(i)}, y_1^{(i)}, y_2^{(i)})$. We can then train the model with maximum likelihood over both the reasoning chain and ranking:

$$\mathcal{L}_{\text{GenRM-Rationalization}}(p_\phi) = \mathbb{E}_{(x, y_1, y_2, r, I) \sim \mathcal{D}}[-\log p_\phi(I|x, y_1, y_2, r) - \log p_\phi(r|x, y_1, y_2)] \tag{8}$$

we refer to this as a post-rationalization approach, similar to the approach described in Section 2.2.2.

**CoT-GenRM-STaR:** Finally we consider an approach where the model self-bootstraps intermediate reasoning using a STaR approach as outlined in Section 2.2.1. We will also consider two loss objectives here—following the filtering strategy described in the above section, we use the standard SFT loss similar to Eq. 8 on reasoning chains that yield the correct judgement. We denote models trained with this objective as STaR-SFT.

Alternatively, we would like to utilize all the sampled data, including reasoning chains that yield wrong judgments. Similar to the reasoning approach in Pang et al. (2024) we create a dataset of preference pairs $\mathcal{D} = \{x^{(i)}, y_1^{(i)}, y_2^{(i)}, r_w^{(i)}, I_w^{(i)}, r_l^{(i)}, I_l^{(i)}\}$, where $r_w$ are rationales that lead to correct rankings $I_w$ and $r_l$ are rationales that lead to incorrect rankings $I_l$. We then use a DPO optimization objective of the form:

$$\mathcal{L}_{\text{GenRM-DPO}}(p_\phi) = \mathbb{E}_{\mathcal{D}} \left[ \log \sigma \left( \beta \log \frac{p_\phi(I_w, r_w|x, y_1, y_2)}{p_{\text{ref}}(I_w, r_w|x, y_1, y_2)} - \beta \log \frac{p_\phi(I_l, r_l|x, y_1, y_2)}{p_{\text{ref}}(I_l, r_l|x, y_1, y_2)} \right) \right]. \tag{9}$$

We denote models trained with this objective as STaR-DPO.

# 5 EXPERIMENTS

In this section we evaluate the performance of our proposed Generative Reward Modelling approaches as compared to classical Bradley-Terry reward models (Bradley & Terry, 1952), a more recent reward variant called PairRM (Jiang et al., 2023), as well as zero-shot RLAIF evaluation. All reward models are based on the LLaMa-3.1 8B Instruct model (Dubey et al., 2024).

We consider two separate training datasets: UltraFeedback (Cui et al., 2023), a large-scale feedback dataset of 61k pairs focusing on general instruction following, and UltraInteract (Yuan et al.,

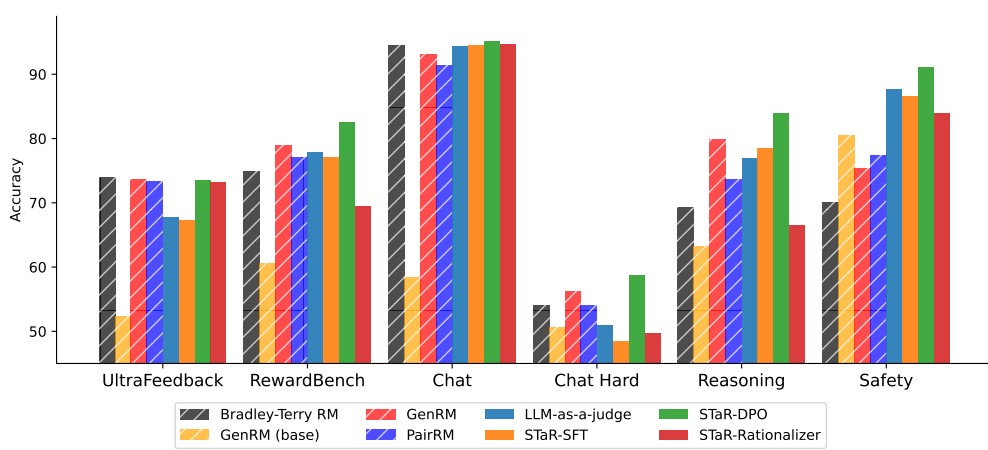

Figure 2: **Comparing generative reward models with prior reward modeling methods** on in-domain (UltraFeedback) data and out-of-domain data (RewardBench). All generative model scores are the result of a majority vote over 32 samples.

2024b), a dataset consisting of multi-turn reasoning trees focusing on math, code and logic, including advanced capabilities such as tool use and environment interaction. We evaluate models on each training dataset as well as on RewardBench (Lambert et al., 2024), a general reward modeling benchmark which consists of four subsets—Chat, Chat (Hard), Reasoning, and Safety. Full training details can be found in Appendix B.

In our experiments we evaluate the following questions:

1. Do generative reward models match the performance of classical reward models?
2. How robust are reward models and how well do they generalize to OOD data?
3. Does reasoning improve reward modelling and can we use inference time compute to improve results?
4. Do we need strong reasoning data to train generative RMs or can we bootstrap the reasoning from the model itself?
5. Does improved reward accuracy performance yield improved policy performance?

### 5.1 PERFORMANCE OF GENERATIVE RMs ON GENERAL ASSISTANT TASKS

We show our first main set of results in Fig. 2. All models are trained on the UltraFeedback dataset and evaluated on a held-out split of in-domain data as well as on RewardBench.

We first evaluate the zero-shot performance of the LLaMa 3.1 8B Instruct model as an evaluator using both CoT prompting with self-consistency (LLM-as-a-judge in Figure 2) and acting as a classifier directly outputting the response ranking (GenRM (base) in 2). We see that using that prompting the model to reason over the answers significantly boosts performance from 52.25% to 67.75% on the UltraFeedback evaluation dataset and from 60.60% to 75.18% accuracy on RewardBench.

However, when we compare the zero-shot methods with trained models, we find that both approaches substantially under-perform the Bradley-Terry RM, PairRM and trained GenRM models which all have comparable accuracies, around 73-74%. We see that the STaR-DPO model also matches these accuracies in-distribution at 73.9%. On the other hand, the STaR-SFT model achieves in-distribution accuracy of only 67.4%, which essentially shows no change of performance compared to the base LLM.

When we evaluate these models on out-of-distribution tasks (RewardBench) we see that STaR-DPO achieves the strongest result, over both the base model prior (81.9% versus 77.8%) and trained RM with the GenRM having the stronger performance at 78.9%. The STaR-DPO model outperforms or matches baselines across all RewardBench categories. On the other hand, the STaR-SFT model

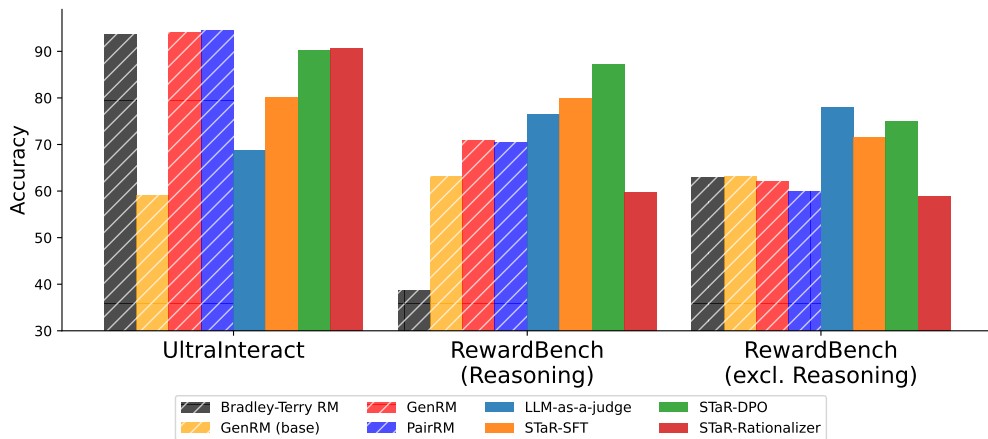

Figure 3: **Comparing generative reward models with prior reward modeling methods** on in-domain (UltraInteract) data and out-of-domain data (RewardBench) split into reasoning and non-reasoning subsets. All generative model scores are the result of a majority vote over 32 samples.

still does not substantially differ from the base model. The GenRM model outperforms the Bradley-Terry RM and PairRM with performance comparable to STaR-DPO across Chat, Chat (Hard) and Reasoning. However, one notable observation is that reasoning-based approaches show significantly stronger performance on the Safety category with the STaR-DPO model achieving 91.0% accuracy versus the best performing PairRM model, which achieves accuracy of 81.8%.

Overall, we see that the STaR-DPO reasoning model matches the best performance on the in-distribution dataset and has the strongest out-of-domain generalization across evaluation categories on RewardBench.

## 5.2 Performance on Reasoning Tasks

We also evaluate performance specifically on reasoning tasks by training models on the UltraInteract dataset, which specifically focuses on challenging evaluations of reasoning chains. The experiments in this section focus on *meta*-reasoning, i.e. the capability to reason about reasoning steps. Figure 3 shows that the STaR-DPO model outperforms STaR-SFT on the UltraInteract evaluation dataset and achieves a significant improvement of 90.2% versus 68.8% for the base model. This is slightly lower than the explicit RM models, which all achieve comparable performance around 94%.

However, we see an interesting divergence on the RewardBench evaluation. For this experiment we show performance on the Reasoning category in RewardBench versus all other categories. We see that the Bradley-Terry RM, PairRM, and GenRM struggle to generalize to the RewardBench data, with the Bradley-Terry model scoring worse than random and the best performing GenRM model achieving 70.8% accuracy, which is worse than the LLM-as-a-judge performance at 76.6%. On the other hand the STaR-DPO significantly outperforms both baselines with 87.2%. This shows that the model successfully generalizes the meta-reasoning capabilities from the training dataset to a different distribution of reasoning prompts and answers. Additionally, on the non-reasoning evaluations in RewardBench, unsurprisingly the LLM-as-a-judge achieves the strongest result with 78.0% accuracy, while all explicit reward models struggle to generalize to these tasks and distributions. At the same time STaR-DPO only suffers a small loss of accuracy on these domains at 75.0%.

Based on the prior experiments we observe that the STaR-DPO model significantly outperforms the base LLM-as-a-judge, but also the GenRM model which does not use reasoning on held out tasks in RewardBench. One major variable is how to generate the reasoning chains used for training. In the experiments described so far rationales were sampled following a STaR-based approach using the same base model. We also evaluate an approach sampling rationales from a post-rationalization model $r \sim q_\phi(r|x, y_1, y_2, I)$, which are then used for SFT training using Eq. 8, we refer to this as STaR-Rationalizer. Results from this approach on UltraFeedback are shown in Fig. 2 and for UltraInteract in Fig. 3. Interestingly we observe that the STaR-Rationalizer model matches the

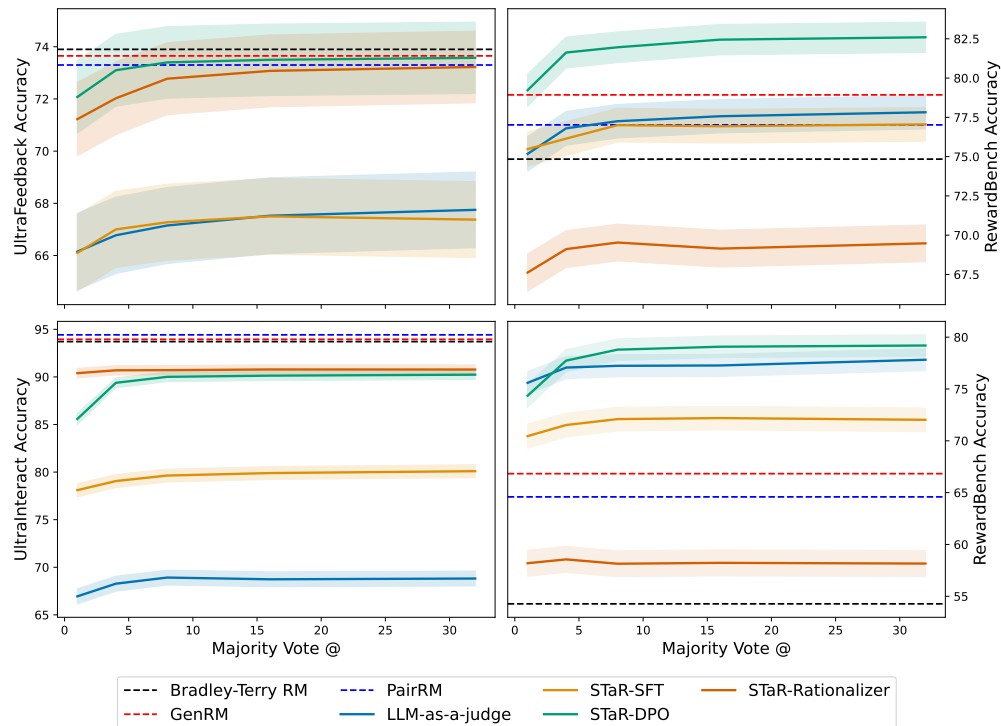

Figure 4: **Comparing generative reward models with prior reward modeling methods** with majority vote evaluation. Top row models are trained on UltraFeedback (Cui et al., 2024), and bottom row models are trained on UltraInteract (Yuan et al., 2024a). Left column models are evaluated on either UltraFeedback or UltraInteract, and right column models are evaluated on RewardBench (Lambert et al., 2024). Solid line methods are sampled to produce final answers and shading reflects a 95% confidence interval.

performance of the STaR-DPO model on both datasets, significantly out-performing the STaR-SFT approach, which uses the same training objective. However, we see that this model struggles to generalize to the RewardBench tasks, under-performing not only the STaR-DPO model, but the base LLM as well on both datasets. This is an interesting empirical phenomenon that warrants further study, but we hypothesize that the core issue is that the training rationales from the post-rationalization model are off-policy for the base model, creating a distribution mismatch during training. While the model is able to learn those rationales on the training distribution it fails on more novel tasks where it generates rationales different from those seen during training.

## 5.3 USING INFERENCE-TIME COMPUTE

In addition to the previously discussed benefits, using LLMs as reward models also allows us to utilize additional inference time compute to improve performance. Indeed our results from the previous sections show that using COT prompting to induce reasoning in the evaluator model can significantly improve performance on new prompts and tasks over the base GenRM approach (which does not use CoT). In this section we show further results on accuracy using self-consistency and majority voting. Our results are shown in Fig. 4. We see that majority voting at 32 improves performance consistently and adds 1.6% accuracy on the UltraFeedback Dataset and 3.8% on RewardBench in that case. On UltraInteract majority voting improves performance by 4.6% and 4.9% on Reward-Bench. This indicates that "System 2" types of reasoning approaches can significantly improve the accuracy of the critic model. We believe using models with strong reasoning to provide feedback and evaluation to other models is a promising direction.

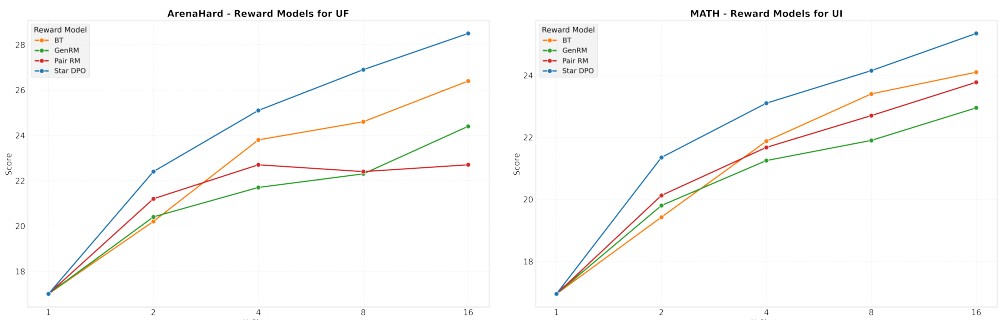

Figure 5: Best-Of-N performance for LLaMa 3.8 Instruct 8b model with different reward models. We use ArenaHard evaluations for reward models trained on UltraFeedback and MATH 500 for models trained on UltraInteract, which is focused on reasoning.

### 5.4 BOOTSTRAPPING RATIONALES

We show additional evaluations in Table 1 using different models to bootstrap reasoning on the UltraFeedback domain. We see similar results to the above when using rationales generated by a strong GPT-4 model, which significantly under-performs the standard STaR methods, however this is alleviated with additional on-policy training. Finally we see that bootstrapping reasoning with samples from the stronger LLaMa 3.1 70B Instruct model from the same family slightly improves performance on UltraFeedback, but leads to somewhat worse generalization to RewardBench. Our results indicate that on-policy training of the critic models can meaningfully impact performance.

| Bootstrap Source | UltraFeedback | RewardBench |
|---|---|---|
| Llama3.1 8B | 68.63 | 77.34 |
| | 67.68 | 77.61 |
| | 67.38 | 77.05 |
| Llama3.1 70B | 70.50 | 77.09 |
| | 70.13 | 68.78 |
| | 69.58 | 63.43 |
| GPT-4 | 62.85 | 69.58 |
| | 68.55 | 75.63 |
| | 71.73 | 78.29 |
| GPT-4 (full) | 62.60 | 70.52 |

Table 1: **Bootstrapping STaR-SFT with models of different capabilities.** All methods train a Llama 3.1 8B model using rejection sampling from the bootstrap source. Only the first iteration of data comes from the bootstrap source. GPT-4 (full) is trained entirely on the reasoning from GPT-4. Scores are majority vote over 32 samples.

### 5.5 BEST-OF-N SAMPLING WITH GENERATIVE RMS

We further evaluate the degree to which improved reward modeling translates to downstream model performance. Due to the high computational cost of RL training models with all baselines we evaluate Best-Of-N performance on top of the existing LLaMA 3 8B Instruct model. For pairwise models, we use the probability of out-performing an answer from some reference model as a scoring function as outlined in Section 3. We evaluate reward models trained on UltraFeedback on the ArenaHard Li et al. (2024) benchmark and reward models trained on UltraInteract, on the MATH 500 Hendrycks et al. (2021) test question set. Results are shown in Fig. 5. While performance improves with more samples across all reward models, the CoT-GenRM with STaR-DPO training shows the most favorable scaling. That is, the reward model's improved generalization performance on the classification task also carries over to improved **policy** in the Best-Of-N setting as well.

### 6 RELATED WORK

The concept of using language models for providing feedback, also known as constitutional AI or RLAIF Bai et al. (2022b) has gained significant traction in recent years. Zheng et al. (2023b) further popularized the paradigm of LLM evaluation ("LLM-as-a-judge"), demonstrating that strong language models can effectively perform judgments with chain-of-thought (CoT) reasoning that approximate human evaluation. Building on this, Kim et al. (2024) proposed Prometheus, employing supervised fine-tuning from a powerful model to provide CoT reasoning and scoring, demonstrating strong evaluation performance from a smaller open model. In the current work we show that

zero-shot LLM evaluations may not fully align with human feedback and significant improvements in accuracy can be gained from additional fine-tuning. Moreover, in our proposed approach of combining RLAIF with STaR-based tuning we do not require ground-truth reasoning or supervision from a stronger model. Concurrently with this work, Zhang et al. (2024) presented Generative Verifiers, training CoT-GenRM with an SFT objective to act as a verifier for mathematical reasoning. They find similiar observations to our experiments in Sections 5.2 and 5.3. However, one notable exception is that unlike our approach they rely on access to full reference solutions at training time. Another concurrent work in this direction is Ankner et al. (2024), which combines CoT reasoning generation with Bradley-Terry reward modelling. They present empirical findings on the benefits of reasoning, similar to the ones we show in Section 5.1, although crucially, they rely on a separate strong model to provide rationales and require the Bradley-Terry reward model hybrid architecture, while we use fully self-bootstrapped rationales in a full language modelling setup without the need for any additional architecture overhang.

A number of concurrent works have also proposed approaches for self-bootstrapping generative critics. In Wang et al. (2024c), the authors present an approach similar to our STaR-SFT method using data augmentation to create synthetic preference pairs. On the other hand Wu et al. (2024) instead uses DPO for optimizing the evaluator, but the feedback is based on additional meta-judge evaluator. Similarly, Wang et al. (2024b) also generates data using a number of augmentation techniques and deploys DPO training. We note that these data generation techniques are complementary to our approach which focuses on using STaR-based methods to allign LLM generative reward models and replace explicit reward models.

## 7 CONCLUSION AND FUTURE DIRECTIONS

To conclude, this work introduces Generative Reward Models (GenRM) as a novel framework that combines the strengths of Reinforcement Learning from Human Feedback (RLHF) and Reinforcement Learning from AI Feedback (RLAIF) to improve preference modeling for large language models (LLMs). By leveraging self-generated reasoning traces and iterative training loops, GenRM can fine-tune LLMs to better align with human preferences, addressing key limitations of both human and AI feedback mechanisms. The GenRM approach demonstrates that integrating chain-of-thought reasoning within preference modeling can significantly improve both in-distribution and out-of-distribution performance compared to baselines. The Chain-of-Thought (CoT) GenRM variant augments preference judgment tasks with intermediate reasoning traces, boosting model performance by encouraging more logical, step-wise decision-making. This leads to better generalization on complex tasks such as reasoning and safety-related scenarios. Empirical results show that GenRM and its variants maintain competitive in-distribution accuracy while outperforming traditional methods on out-of-distribution tasks. Notably, STaR-DPO models, which rely on reasoning-based preference optimization, demonstrate superior robustness and performance across benchmarks. In **summary**, Generative Reward Models present a significant advancement in combining human and AI feedback to enhance the quality of synthetic preference labels. It improves scalability, out-of-distribution generalization, and model performance. Future exploration into online optimization, robust reasoning, and adaptation to multimodal tasks will be key to realizing the full potential of this framework in practical deployments.

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

# A    EVALUATION EXAMPLE

**Prompt**

User: Name two animal species that live in the ocean.
[The Start of Assistant A's Answer]
Dolphin and shark.
[The End of Assistant A's Answer]
[The Start of Assistant B's Answer]
Common ocean animals include sharks, whales, and dolphins.
[The End of Assistant B's Answer]

**STaR-DPO**

For the user's question, which is to name two animal species that live in the ocean, we have responses from two assistants.

Assistant A's response is "Dolphin and shark." Although it names two species that live in the ocean, it lacks depth and detail. The response does not provide further information about the species, and it only mentions two examples.

Assistant B's response is "Common ocean animals include sharks, whales, and dolphins." This response is more detailed and provides additional information about the types of ocean animals. However, the question specifically asks for two species, and Assistant B's response names three species.

Considering the user's question, I would say that **Assistant A's response is more suitable because it directly answers the question by naming two animal species** that live in the ocean.

[[A]]

**LLM-as-a-judge**

To compare the two responses, let's consider the factors mentioned earlier.

Helpfulness: Both responses provide the user with two (in the case of Assistant A) or more (in the case of Assistant B) ocean-dwelling animal species. However, Assistant B provides a **more comprehensive list of common ocean animals**, which could be more helpful to the user.

Relevance: Both responses are directly related to the user's question about ocean-dwelling animal species.

Accuracy: Both responses are accurate in providing examples of ocean-dwelling animals. However, Assistant B provides a more specific list of common ocean animals.

Depth: Assistant B's response is more detailed, providing a list of common ocean animals, while Assistant A's response is more concise but less detailed.

Creativity: Both responses are straightforward and do not show much creativity.

Level of detail: Assistant B's response provides more detail by listing specific ocean animals, while Assistant A's response is more general.

Overall, considering the factors above, **Assistant B's response provides more helpful and detailed information**, making it a better choice for the user's question.

[[B]]

Figure 6: An example where LLM-as-a-judge fails to provide an accurate judgement, but STaR-DPO succeeds. Sentences where critical reasoning takes place are bolded for emphasis.

# B    ADDITIONAL EXPERIMENT DETAILS

## B.1    PROMPTS

Our prompts follow Zheng et al. (2023a) with the following differences. For our LLM-as-a-judge and STaR (Figure 7), we remove the tie. For GenRM (Figure 9), we remove the brackets, and remove the explanation request in the system prompt. For Rationalizer (Figure 8), we remove the output instructions from the system prompt, and add a prompt to explain why A or B is better.

All prompts are using llama 3.1 instruct chat templates to format the prompts

**System:**
Please act as an impartial judge and evaluate the quality of the responses provided by two AI assistants to the user question displayed below. You should choose the assistant that follows the user's instructions and answers the user's question better. Your evaluation should consider factors such as the helpfulness, relevance, accuracy, depth, creativity, and level of detail of their responses. Begin your evaluation by comparing the two responses and provide a short explanation. Avoid any position biases and ensure that the order in which the responses were presented does not influence your decision. Do not allow the length of the responses to influence your evaluation. Do not favor certain names of the assistants. Be as objective as possible. After providing your explanation, output your final verdict by strictly following this format: "[[A]]" if assistant A is better, "[[B]]" if assistant B is better

**User:**
[Chat Context]
{chat}

[The Start of Assistant A's Answer]
{answer_a}
[The End of Assistant A's Answer]

[The Start of Assistant B's Answer]
{answer_b}
[The End of Assistant B's Answer]

Figure 7: Prompt structure for LLM-as-a-Judge and STaR-SFT/DPO methods

**System:**
Please act as an impartial judge and evaluate the quality of the responses provided by two AI assistants to the user question displayed below. You should choose the assistant that follows the user's instructions and answers the user's question better. Your evaluation should consider factors such as the helpfulness, relevance, accuracy, depth, creativity, and level of detail of their responses. Begin your evaluation by comparing the two responses and provide a short explanation. Avoid any position biases and ensure that the order in which the responses were presented does not influence your decision. Do not allow the length of the responses to influence your evaluation. Do not favor certain names of the assistants. Be as objective as possible.

**User:**
[Chat Context]
{chat}

[The Start of Assistant A's Answer]
{answer_a}
[The End of Assistant A's Answer]

[The Start of Assistant B's Answer]
{answer_b}
[The End of Assistant B's Answer]

Explain why response (A/B) is better than response (B/A).

Figure 8: Prompt structure for generating rationals for rationalizer methods

---

**System:** Please act as an impartial judge and evaluate the quality of the responses provided by two AI assistants to the user question displayed below. You should choose the assistant that follows the user's instructions and answers the user's question better. Your evaluation should consider factors such as the helpfulness, relevance, accuracy, depth, creativity, and level of detail of their responses. Avoid any position biases and ensure that the order in which the responses were presented does not influence your decision. Do not allow the length of the responses to influence your evaluation. Do not favor certain names of the assistants. Be as objective as possible. Output your verdict by strictly following this format: "A" if assistant A is better, "B" if assistant B is better

**User:**
[Chat Context]
{chat}

[The Start of Assistant A's Answer]
{answer_a}
[The End of Assistant A's Answer]

[The Start of Assistant B's Answer]
{answer_b}
[The End of Assistant B's Answer]

---

Figure 9: Prompt structure for GenRM methods

## B.2 HYPERPARAMETERS

### B.2.1 TRAINING HYPERPARAMETERS

For all training, we use the GPT-NeoX framework (Andonian et al., 2023). Models use the same hyperparameters across each dataset. Table 2 contains our hyperparameter choices for each training method.

| Model | AdamW LR | AdamW $(\beta_1, \beta_2)$ | Optimizer Weight Decay | Optimizer Schedule | LR warmup | $\beta$ |
|-------|----------|--------------------|------------------------|--------------------|-----------|---------|
| STaR-SFT | 1.0e-6 | | | | | n/a |
| STaR-DPO | 1.0e-6 | | | | | 1.0 |
| STaR-SFT Rationalizer | 2.0e-5 | | | | | n/a |
| STaR-IPO Rationalizer | 1.0e-6 | (0.9, 0.95) | 0.1 | cosine | 0.1 | 0.4 |
| Bradley-Terry RM | 1.0e-6 | | | | | n/a |
| GenRM | 1.0e-6 | | | | | n/a |
| PairRM | 1.0e-6 | | | | | n/a |

Table 2: Hyperparameters for different models.

### B.2.2 GENERATION HYPERPARMETERS

All models use the following settings for generation:

- SGLang version: 0.3.0 (Zheng et al., 2024)
- Temperature: 1.0
- Top-p: 0.95

## B.3 STAR ITERATIONS

For each training run, we only sample from each pairwise data point one time. We split each dataset into three equal portions, and then apply the model to one of these splits without reusing any splits. We do not share any data between iterations, each iteration is fully sampled from the split, and does not include any of the previously generated data into the training. To generate the new data points, we sample from the latest model on the current split. From there, we train the latest model on that

split in an online manner. In order to accomplish roughly the same number of training steps for each dataset, ultrafeedback uses 3 epochs from this generated data, while ultrainteract uses 1 epoch on this data.

## C    RESULTS ACROSS ITERATIONS

| Training Dataset | UltraFeedback | | | | UltraInteract | | | |
|---|---|---|---|---|---|---|---|---|
| Evaluation Dataset | UltraFeedback | | RewardBench | | UltraInteract | | RewardBench | |
| **Method** (iteration) | Maj@1 | Maj@32 | Maj@1 | Maj@32 | Maj@1 | Maj@32 | Maj@1 | Maj@32 |
| STaR-SFT (1) | 67.42 | 68.62 | 76.10 | 77.34 | 72.33 | 75.03 | 72.33 | 74.20 |
| STaR-SFT (2) | 67.05 | 67.67 | 75.71 | 77.60 | 76.26 | 79.18 | 72.95 | 74.52 |
| STaR-SFT (3) | 66.10 | 67.38 | 75.48 | 77.05 | 78.10 | 80.10 | 70.45 | 72.03 |
| STaR-DPO (1) | 69.30 | 71.28 | 78.29 | 81.27 | 82.23 | 86.46 | 73.13 | 78.04 |
| STaR-DPO (2) | 71.70 | 72.98 | 78.46 | 81.94 | 84.70 | 88.40 | 76.31 | 79.92 |
| STaR-DPO (3) | 72.08 | 73.58 | 79.23 | 82.60 | 85.58 | 90.23 | 74.36 | 79.20 |
| Rationalizer (1) | 70.90 | 73.05 | 71.73 | 75.83 | 84.31 | 85.11 | 67.04 | 68.61 |
| Rationalizer (2) | 71.05 | 73.62 | 68.54 | 71.91 | 88.19 | 88.55 | 60.28 | 61.37 |
| Rationalizer (3) | 71.23 | 73.22 | 67.62 | 69.48 | 90.39 | 90.77 | 58.19 | 58.16 |

Table 3: STaR method evaluation results throughout training iterations.

## D    NUMERICAL RESULTS

| | UltraFeedback | RewardBench | Chat | Chat Hard | Reasoning | Safety |
|---|---|---|---|---|---|---|
| Bradley-Terry RM | 73.90 | 74.84 | 94.41 | 53.95 | 69.26 | 70.12 |
| GenRM (base) | 52.25 | 60.60 | 58.38 | 50.66 | 63.15 | 80.44 |
| GenRM | 73.65 | 78.93 | 93.02 | 56.14 | 79.82 | 75.38 |
| PairRM | 73.30 | 77.02 | 91.34 | 53.95 | 73.59 | 77.29 |
| LLM-as-a-judge | 67.75 | 77.82 | 94.33 | 50.95 | 77.02 | 87.72 |
| STaR-SFT | 67.38 | 77.05 | 94.47 | 48.40 | 78.44 | 86.56 |
| STaR-DPO | 73.58 | 82.60 | 95.19 | 58.71 | 83.91 | 91.06 |
| STaR-Rationalizer | 73.22 | 69.48 | 94.62 | 49.71 | 66.50 | 83.88 |

Table 4: **Comparing generative reward models with prior reward modeling methods** on in-domain (UltraFeedback) data and out-of-domain data (RewardBench and all subsets). All generative model scores are the result of a majority vote over 32 samples.

| | UltraInteract | RewardBench | Chat | Chat Hard | Reasoning | Safety |
|---|---|---|---|---|---|---|
| Bradley-Terry RM | 93.71 | 56.77 | 85.20 | 49.12 | 38.59 | 54.15 |
| GenRM (base) | 59.05 | 31.16 | 58.38 | 50.66 | 63.15 | 80.44 |
| GenRM | 93.95 | 64.22 | 87.99 | 48.25 | 70.84 | 49.80 |
| PairRM | 94.42 | 62.52 | 88.27 | 48.03 | 70.48 | 43.28 |
| LLM-as-a-judge | 68.82 | 77.63 | 94.33 | 51.51 | 76.62 | 88.06 |
| STaR-SFT | 80.10 | 73.70 | 92.35 | 46.07 | 80.02 | 76.37 |
| STaR-DPO | 90.23 | 78.02 | 93.64 | 51.82 | 87.19 | 79.44 |
| STaR-Rationalizer | 90.77 | 59.14 | 82.02 | 42.05 | 59.83 | 52.66 |

Table 5: **Comparing generative reward models with prior reward modeling methods** on in-domain (UltraInteract) data and out-of-domain data (RewardBench and all subsets) split into reasoning and non-reasoning subsets. All generative model scores are the result of a majority vote over 32 samples.

