# OpenReview forum: "Generative Reward Models"
_ICLR.cc/2025/Conference — Submitted to ICLR 2025_

### Official Review · Reviewer_4sEX · 2024-10-29

**Soundness:** 2
**Presentation:** 2
**Contribution:** 2
**Rating:** 5
**Confidence:** 3

**Summary:**

This paper proposes a Generative Reward Modeling (GenRM) approach that combines the main ideas of RLHF and RLAIF, aiming at fine-tuning LLMs with self-generated reasoning traces to generate synthetic preference labels that match human preference judgments. Experiments on Llama 3.1 8B confirm the effectiveness of the proposed method.

**Strengths:**

This research topic is interesting.

**Weaknesses:**

1. Mixing RLHF and RLAIF gets GenRM both their advantages, but also inherits both their disadvantages, and the paper lacks a discussion of that. And the idea doesn't seem very novel.

2. The foundation model in the experiment only uses Llama3.1 8B, which is not quite demonstrating the generality and robustness of the method. Experimenting on a more variety (e.g., mistral series) and size (e.g., 13B, 70B) of models would make this paper more solid.

3. Why does this method perform better on out-of-distribution (OOD) data? It's not very intuitive, not very easy to understand, and it would be better if there was more discussion and analysis of it.

**Questions:**

1. Mixing RLHF and RLAIF gets GenRM both their advantages, but also inherits both their disadvantages. Would you mind analyzing it a little more?

2. The foundation model in the experiment only uses Llama3.1 8B, which is not quite demonstrating the generality and robustness of the method. Experimenting on a more variety (e.g., mistral series) and size (e.g., 13B, 70B) of models would make this paper more solid.

3. Why does this method perform better on out-of-distribution (OOD) data? It's not very intuitive, not very easy to understand, and it would be better if there was more discussion and analysis of it.

---

> ### Author Response · Authors · 2024-11-27
> **Thank you for your feedback**
>
> **Mixing RLHF and RLAIF gets GenRM both their advantages, but also inherits both their disadvantages, and the paper lacks a discussion of that. And the idea doesn't seem very novel.**
>
> Can the reviewer provide some examples of the mentioned disadvantages? In our experiments the proposed approach achieves comparable accuracy on in-distribution data and meaningfully better generalization on OOD data as compared to both fully RLAIF approaches and RLHF reward models. Moreover, we have added additional experiments on ArenaHard and MATH500 (consult the new Figure 5 and Section 5.5 for full results) showing that the improved accuracy translates to better performance when using LLaMa 3 Instruct 8B model (which has already been heavily RLHF-trained) with Best-Of-N sampling.
>
> There have been a few concurrent works that propose similar approaches to our work (differences are more extensively discussed in the Related Works section), but as far as we are aware specifically training RLAIF approaches to align with human (RLHF) data is novel.
>
>
> **The foundation model in the experiment only uses Llama3.1 8B, which is not quite demonstrating the generality and robustness of the method. Experimenting on a more variety (e.g., mistral series) and size (e.g., 13B, 70B) of models would make this paper more solid.**
>
> We evaluate 8 approaches across two large preference datasets. The. LLaMa 3.1. 8B model is considered frontier for its parameter class and we have no reason to believe these results are not transferable to other models or larger parameter classes. Additionally, the requested experiments would require nearly 20,000 GPU-hours on H100 cards.
>
> **Why does this method perform better on out-of-distribution (OOD) data? It's not very intuitive, not very easy to understand, and it would be better if there was more discussion and analysis of it.**
>
> We believe our proposed approach generalizes better than explicit reward modelling (Bradley-Terry and PairRM baselines) because it leverages the underlying LLM’s generative capabilities rather than the discriminative nature of the standard reward models as they are trained on orders of magnitude less data. The question of why reasoning approaches generalize better or are more robust, as well as the dynamics of that process are quite deep and we cannot answer them within the confines of this paper.

---

> > ### Comment · Reviewer_4sEX · 2024-11-29
> >
> > Thanks for the reply, I saw all the comments and I decided to keep my original score.

---

### Official Review · Reviewer_yvkW · 2024-10-31

**Soundness:** 2
**Presentation:** 2
**Contribution:** 2
**Rating:** 3
**Confidence:** 4

**Summary:**

This paper tackles the problem that feedback from current RLAIF methods does not align with human preferences. It introduces a framework named GenRM, which integrates the strengths of RLHF and RLAIF. The authors utilize self-generated reasoning traces and iterative training loops, achieving results on in-distribution tasks that are comparable to Bradley-Terry models and outperforming them on out-of-distribution tasks. Additionally, GenRM surpasses the performance of using LLMs as judges for both types of tasks.

**Strengths:**

The paper presents an iterative framework called GenRM designed to train an LLM using self-generated reasoning traces. This framework is novel to some extent and has been validated on both in-distribution and out-of-distribution tasks through various ablation settings. It demonstrates scalability and good efficiency. The experimental results indicate that integrating chain-of-thought reasoning within preference modeling enhances the model's reasoning ability, contributing valuable insights for future studies in RLAIF and RLHF.

**Weaknesses:**

1. The evaluation of RM is inadequate, lacking experiments that assess agreement with human preferences. Additionally, only one out-of-distribution dataset, RewardBench, is utilized for evaluation.
2. The qualitative results presented in Figure 3 are unconvincing. Both STaR-DPO and LLM-as-a-judge use the same system prompt, as noted in Appendix A.1, which states that evaluations should consider factors such as helpfulness, relevance, accuracy, depth, creativity, and detail. However, your explanation for why STaR-DPO performs better than LLM-as-a-judge suggests that you prioritize instruction following over depth, helpfulness, and detail. While STaR-DPO includes an additional animal species, which is more detailed and helpful, it does not strictly follow the instructions. Furthermore, the system prompt is not always appropriate, as it can sometimes be contradictory.
3. The paper's title is somewhat unclear, making it difficult to discern your contribution. Additionally, some model settings are not sufficiently clear, such as the differences between GenRM and the STaR-SFT or STaR-DPO models.

**Questions:**

1. Are STaR-SFT and STaR-DPO both trained on the same base model, rather than STaR-DPO being trained on STaR-SFT?
2. Where does the RLHF component fit into your framework? I see that you mention "incorporates human feedback in initial stages" in Section 7, but what exactly do you define as the initial stage, and how is human feedback utilized within this stage?
3. Could you provide a detailed summary of all the model settings, including the eight models depicted in Figure 2? Specifically, what is the base model, if it has been trained, what the training objective is, and what the dataset construction process entails? Thank you.

---

> ### Author Response · Authors · 2024-11-27
> **Thank you for your feedback!**
>
> **The evaluation of RM is inadequate, lacking experiments that assess agreement with human preferences. Additionally, only one out-of-distribution dataset, RewardBench, is utilized for evaluation.**
>
> In our experiments we utilize the UltraFeedback and UltraInteract preference datasets, which are currently standard for preference modelling in the literature. Moreover, existing human preference datasets, such as TLDR and the Anthropic Helpful and Harmless are more easily solved by existing models and are significantly easier than UF and UI, which still remain a challenge for smaller models and even advanced non-specialized models.
>
> We would also like to point out that RewardBench has become the standard benchmark for reward modelling with over 100 frontier reward models evaluated on the official leaderboard and covers multiple datasets with different use cases - general chat/instruction-following, reasoning and safety.
>
> **The qualitative results presented in Figure 3 are unconvincing. Both STaR-DPO and LLM-as-a-judge use the same system prompt, as noted in Appendix A.1, which states that evaluations should consider factors such as helpfulness, relevance, accuracy, depth, creativity, and detail. However, your explanation for why STaR-DPO performs better than LLM-as-a-judge suggests that you prioritize instruction following over depth, helpfulness, and detail. While STaR-DPO includes an additional animal species, which is more detailed and helpful, it does not strictly follow the instructions. Furthermore, the system prompt is not always appropriate, as it can sometimes be contradictory.**
>
> We should note that these models are acting as judges only and do not generate responses to the query. That is they are only evaluating the answers from the generator model. Their chains-of-thought serve to justify their rankings and are thus **evaluated only for accuracy over the preference datasets**.
>
> We use standard LLM-as-a-judge prompts from MTBench that have been established in the RLAIF literature to ensure that our experimental results reflect a fair evaluation of different methods.
>
>
> **The paper's title is somewhat unclear, making it difficult to discern your contribution. Additionally, some model settings are not sufficiently clear, such as the differences between GenRM and the STaR-SFT or STaR-DPO models.**
>
> The contribution of the current work is to propose and evaluate algorithms to specifically train models to act as evaluators in RLHF training and align their judgements with a preference distribution. In essence we train LLMs to act as generative critics (reward models) for RLHF.
> STaR-SFT and STaR-DPO are two different objectives for training the Generative Reward Model (GenRM) as outlined in Section 4.
>
>
> **Are STaR-SFT and STaR-DPO both trained on the same base model, rather than STaR-DPO being trained on STaR-SFT?**
> These are both trained on the same reference model. These are different objectives of training the Generative Reward Model as outlined in Section 4 with experiential details in Section 5 and 5.1.
>
>
> **Where does the RLHF component fit into your framework? I see that you mention "incorporates human feedback in initial stages" in Section 7, but what exactly do you define as the initial stage, and how is human feedback utilized within this stage?**
>
> The initial stage of standard RLHF pipelines is obtaining a reward model. We explicitly suggest how to use an LLM (RLAIF) and align its preferences with the **human preference distribution**. The LLM is then used as a judge to provide preference feedback for the policy evaluation or training.
>
> **Could you provide a detailed summary of all the model settings, including the eight models depicted in Figure 2? Specifically, what is the base model, if it has been trained, what the training objective is, and what the dataset construction process entails? Thank you.**
>
> All judge models use the same baseline LLaMa 3.1 Instruct 8B model. For the Best-Of-N experiments we use the slightly weaker LLaMa 3 Instruct 8B model, since LLaMa 3.1 seems to mostly saturate the considered benchmarks in its parameter class due to its more extensive post-training.
> The training approaches we evaluate are described in Section 4 (GenRM, STaR-SFT, STaR-DPO, STaR-Rationalizer). The Bradley-Terry RM (of which PairRM is a variant) and LLM-as-a-judge baselines are described in the Preliminaries section.

---

> > ### Comment · Reviewer_yvkW · 2024-11-30
> >
> > Thanks for the clarification! After looking through other reviewers' comments, I decided to keep my score.

---

### Official Review · Reviewer_Xwva · 2024-11-02

**Soundness:** 2
**Presentation:** 2
**Contribution:** 3
**Rating:** 5
**Confidence:** 3

**Summary:**

This paper proposes a reward modeling formulation GenRM that explicitly indicates the pairwise preference with $\pi_{\phi}(I | x, y_1, y_2)$, which allows a natural combination of RLHF and RLAIF approaches. By incorporating 1) CoT of Self-taught Reasoning traces of rationalization models, and 2) DPO-style optimization objective, the trained model STaR-DPO achieves on-par performance with Bradley-Terry reward models on IID settings and generalizes much better on OOD settings, e.g. safety and reasoning tasks.

**Strengths:**

The strength of this paper is its potential significance and novelty. The proposed direct formulation of the preference indicator looks interesting to me, which may allow more types of approaches and data sources to be combined to train stronger reward models, especially in real-world scenarios that require strong generalization ability.

**Weaknesses:**

The clarity and quality of the paper can be significantly improved. Specifically, for clarity, the organization of the experimental section can be re-structured to highlight the answers for the proposed questions in lines 296-302. Also, the accuracy numbers in Figure 2 and Figure 4 are recommended to be included in the paper, at least in the Appendix.

Concerning quality, it would be strongly recommended to have policy models trained on STaR-DPO to see whether this improvement in reward modeling can truly translate into gains in better policy models, such as better instruction-following/math/code LLMs. The detailed comments are listed as follows.

### Detailed comments
  * [Clarity] Section 4: Pseudo-code of the proposed algorithm is recommended, at least for STaR-DPO.
  * [Clarity] Section 5: it is recommended to provide separated tables/figures to highlight each point in lines 296-302. Some additional rewards that explicitly answer the question can greatly improve clarity.
  * **[Clarity]** line 306-323: the analysis here can be organized and summarized in several important remarks, especially for those relevant to the questions in lines 296-302. Others can be moved to the appendix.
  * [Clarity] line 224: typo, there is a redundant comma ","
  * [Clarity] line 235": typo, "fine tuning" -> "fine-tuning"
  * [Clarity] line 471: "by a a" -> "by a"
  * [Clarity] line 234, 244, 258: As $\pi$ normally indicates policies in RL, it would be recommended to change this with another symbol.

  * [Quality, Clarity] Section 5, Figures 2 & 4: better include the specific number of accuracies in Tables.
  * [Quality, Clarity] Section 5: a table that explicitly states the gain of the utilized techniques would be strongly encouraged, including GenRM formulation, CoT with other models, CoT with Self-Taught Reasoning, DPO, and test-time computing.
  * **[Quality]** It is strongly encouraged to include results that show the improvements in policy models trained with STaR-DPO.

**Questions:**

I am wondering if there are results showing the improvements of policy models trained on STaR-DPO.

---

> ### Author Response · Authors · 2024-11-27
>
> We thank the reviewer for the detailed comments on the clarity and presentation of our work, we have incorporated their comment into our manuscript with several typo/style fixes, new experiments and full numerical tables in the Appendix.
>
> We propose and evaluate a large number of design choices and ablations and it would be difficult and significantly computationally expensive to also train policies for each proposed experiment. Instead we have used existing instruction-tuned and RLHF models with further Best-Of-N inference-time sampling experiments using the proposed reward models on ArenaHard and Math500. In summary, we see that the better generalization performance of GenRM-CoT translates to better downstream policy performance as well. Please consult the updated Figure 5 and Section 5.5 for full results.

---

> > ### Comment · Reviewer_Xwva · 2024-11-29
> > **Thanks for the clarification**
> >
> > Thanks for the clarification. This paper has high potential and can be improved for the better. I would like to keep my current score.

---

### Official Review · Reviewer_674z · 2024-11-04

**Soundness:** 2
**Presentation:** 3
**Contribution:** 2
**Rating:** 5
**Confidence:** 4

**Summary:**

* The paper proposes GenRM (no CoT), which is similar to PairRM [1], focusing on direct pairwise preference prediction without the Bradley-Terry assumption. The core technical contribution is mainly about removing the explicit reward modeling design.
* Introduces CoT-GenRM (similar to Self-Rewarding LMs [2]) with two training approaches: bootstrapping and rationalization. Through experimental analysis, the authors found that while rationalization works well in-domain, it doesn't generalize effectively to OOD domains. The paper demonstrates this through comprehensive evaluations.
* A key advantage demonstrated is that self-consistency can be used as test-time compute scaling to improve preference modeling accuracy. The experimental results show that majority voting with 32 samples can significantly boost model performance across different evaluation settings.

[1] PairRM: https://arxiv.org/abs/2306.02561
[2] Self-Rewarding LMs: https://arxiv.org/abs/2401.10020

**Strengths:**

* The paper successfully demonstrates that majority voting (self-consistency) at test time can substantially improve preference modeling accuracy. The results show consistent improvements across different datasets, providing a practical way to enhance preference modeling through additional compute at inference time.
* The evaluation for reward modeling covers both in-domain (UltraFeedback) and out-of-domain (RewardBench) datasets. The experimental results demonstrate robust performance improvements, particularly in OOD scenarios where traditional approaches often struggle.
* The paper provides strong empirical evidence that DPO training works effectively with CoT-GenRM. The experiments show that this training approach leads to better generalization compared to other training methods.

**Weaknesses:**

* The evaluation is limited in demonstrating what the preference accuracy improvements mean for actual policy performance. While the paper shows improvements in preference modeling accuracy, this metric heavily depends on the policy sample distribution. Notably absent is an analysis of Best-of-N (BoN) performance with the proposed reward model, which would be crucial for understanding practical impact.
* The practical implementation raises significant concerns. The requirement for 32 majority votes and the limitation to pairwise comparison settings could make it challenging to implement in real-world scenarios, particularly in PPO. The paper doesn't adequately address how to handle these computational requirements in practice.
* The core concepts of GenRM (no CoT) and CoT-RM have been previously explored in literature, such as PairRM [1] and Self-Rewarding LMs [2]. The paper would benefit from more thorough comparisons with existing methods to better establish its novel contributions.

**Questions:**

* Can you provide experimental results showing the Best-of-N performance with your reward model? This would help understand how the improvements in preference modeling translate to actual policy performance.
* Given the computational requirements of majority voting, how would you recommend implementing this approach practically in PPO? What modifications or optimizations might be needed?
* Could you elaborate on the comparisons with similar existing methods in the literature? Specifically, how does your approach differ from and improve upon previous work like PairRM [1] and Self-Rewarding LMs [2]?

---

> ### Author Response · Authors · 2024-11-27
>
> We would like to thank the reviewer for the detailed feedback!
>
> **The evaluation is limited in demonstrating what the preference accuracy improvements mean for actual policy performance. While the paper shows improvements in preference modeling accuracy, this metric heavily depends on the policy sample distribution. Notably absent is an analysis of Best-of-N (BoN) performance with the proposed reward model, which would be crucial for understanding practical impact.**
>
> We have added a set of evaluations in the Appendix for Best-Of-N using different reward model approaches on MTBench, ArenaHard and Math500 showing that the generative reward model with chain-of-thought’s higher classification accuracy translates to better downstream performance as well. Please consult Section 5.5 for the full updated results.
>
>
> **The practical implementation raises significant concerns. The requirement for 32 majority votes and the limitation to pairwise comparison settings could make it challenging to implement in real-world scenarios, particularly in PPO. The paper doesn't adequately address how to handle these computational requirements in practice.**
>
> We describe the connection between the classical RLHF objective and preference modelling in Section 3. In particular classical PPO pipelines still optimize the probability that the generated answer out-performs some reference source distribution. Under this framework, we can directly substitute with the pairwise preference likelihood obtained from a GenRM model.
>
> It is true that our approach requires a higher amount of inference time compute, similar to all RLAIF approaches. While this is indeed a limitation, we believe that inference-time techniques will lead to significant capabilities improvements going forward.
>
>
>
> **The core concepts of GenRM (no CoT) and CoT-RM have been previously explored in literature, such as PairRM [1] and Self-Rewarding LMs [2]. The paper would benefit from more thorough comparisons with existing methods to better establish its novel contributions.**
>
> Our key suggestion is to leverage the existing LLM capabilities to directly model preferences (rewards) in natural language. While PairRM is also a pairwise preference model it still uses classical reward parameterization and training formulations (instead of natural language).
>
> Our approach is a form of RLAIF, which the mentioned work, Self-Rewarding LMs belongs to. However prior RLAIF approaches have leveraged the model’s already existing capabilities to act as a judge (LLM-as-a-judge baseline in our paper). **Instead, we suggest explicitly training a model to act as a judge.** We propose an algorithm to align the judge model with datasets of human preferences using self-generated reasoning chains and evaluate a slew of design choices empirically, which we believe is a novel contribution.
>
>
> Can you provide experimental results showing the Best-of-N performance with your reward model? This would help understand how the improvements in preference modeling translate to actual policy performance.
>
> We have updated our manuscript with examples on ArenaHard and Math 500, please consult the updated Section 5.5.
>
>
> **Given the computational requirements of majority voting, how would you recommend implementing this approach practically in PPO? What modifications or optimizations might be needed?**
>
> The core idea of our approach is to specifically train an evaluation model to align with human preferences and act as a judge. In this framework, the trained model is fully compatible with existing RLAIF pipelines, including approaches outlined in Section 3.
>
> **Could you elaborate on the comparisons with similar existing methods in the literature? Specifically, how does your approach differ from and improve upon previous work like PairRM [1] and Self-Rewarding LMs [2]?**
>
> Please see above, we will expand our related works section to include that discussion.

---

### Meta-Review · Area_Chair_yP5B · 2024-12-20

**Metareview:**

The main goal of this paper is to propose a hybrid approach that combines RLHF and RLAIF approaches for reward modeling. The proposed method, STaR-DPO, shows on-par performance with Bradley-Terry reward models under IID settings and better generalization under OOD settings. Reviewers have significant concerns about the novelty, as pointed out by Reviewer 674z, and the practicality of the proposed model, given that it needs 32 models for majority voting. Reviewers also raised concerns about the clarity of the paper as well as the experimental setup. The authors are encouraged to incorporate these comments when preparing the next iteration of the paper.

**Additional Comments On Reviewer Discussion:**

In consensus of rejection.

---

### Decision · Program_Chairs · 2025-01-22

Reject